# Trading Complexity for Expressivity Through Structured Generalized Linear Token Mixing

**Erwan Fagnou** [1]  **Paul Caillon** [1]  **Blaise Delattre** [1,2]  **Alexandre Allauzen** [1,3]

## Abstract

Token mixing layers play a key role in how language models can learn and generate long-range dependencies. Their efficiency relies on the necessary trade-off between decoding speed and the memory requirements, along with the cache size. Considering causal generation, this paper explores new trade-offs thanks to a unified framework which separates two crucial features: (i) the direct influence of inputs on outputs in one generation step; (ii) the recurrent propagation of information through past outputs. This framework encompasses major architectures such as attention and state-space models, but also generalizes the recurrence equations by allowing each state to depend on multiple past states rather than only the immediate predecessor. By introducing structure, we design new recurrence patterns that provably achieve the desired complexity, while providing theoretical insights on their expressivity – trading runtime for expressivity in a principled way. Empirical validation is performed on synthetic tasks, along with language modeling. Together, these results provide a unified toolkit for the understanding and design of efficient and expressive token mixers across model families.

## 1. Introduction

Token mixing is the mechanism by which sequence models exchange information across tokens or positions. For decades now, the design of modern architectures has explored how to address this key challenge. For instance, early recurrent neural networks (RNNs) propagate information

between two consecutive tokens through a hidden vector, which is updated iteratively through the sequence. They however suffer from multiple flaws: sequentiality limits parallelism, long-range dependencies are difficult to capture, and the optimization is a burden. Transformers replaced recurrence with self-attention, enabling global one-hop interactions and massive parallelism during the training phase. This architecture quickly became the backbone of large language models (Vaswani et al., 2017; Devlin et al., 2019; Brown et al., 2020). Yet their quadratic cost in sequence length remains a bottleneck as context windows scale toward hundreds of thousands or even millions of tokens.

In response, alternative kinds of mixers have been explored in the field. For instance, low-rank and kernelized forms of linear attention reduce the cost of the softmax operator (Katharopoulos et al., 2020; Choromanski et al., 2021; Wang et al., 2020). In a different way, state space models (SSMs) reinterpret token mixing as structured linear recurrences, with certain formulations (Gu et al., 2022c) reducible to fast convolutions and others designed for recurrent or scan-friendly execution (Gu & Dao, 2024; Dao & Gu, 2024). These last versions can achieve strong performance on long-range benchmarks. More recently, hybrid models combine these mechanisms – mixing SSMs with local or sparse attention, or alternating different mixer types across layers – to balance expressivity, efficiency, and cache size (De et al., 2024; Zancato et al., 2024; NVIDIA et al., 2025; Team et al., 2025a; Secrieru et al., 2025).

This growing diversity underscores a key challenge: token mixing is no longer embodied by a single operator but by a toolbox of mechanisms, each making distinct trade-offs between complexity, expressivity, and execution mode. One underexplored but important dimension is the order of recurrence: whereas classical RNNs and most SSMs propagate information through a single previous state, higher-order recurrences extend the dependence to multiple past states. Expressivity is clearly improved but at the cost of an extra-complexity. Although this idea was previously overlooked, a few notable efforts include log-linear attention (Guo et al., 2026), which induces a logarithmic-order recurrence, Chimera (Lahoti et al., 2025) which generalizes the SSM recurrence equation to graphs, and ChaCAL (Fag-

[1]Miles Team, LAMSADE, Université Paris Dauphine-PSL, Paris, France [2]Department of Computer Science, School of Computing, Institute of Science Tokyo, Tokyo, Japan [3]ESPCI PSL, Paris, France. Correspondence to: Erwan Fagnou <erwan.fagnou@dauphine.psl.eu>.

*Proceedings of the $43^{rd}$ International Conference on Machine Learning*, Seoul, South Korea. PMLR 306, 2026. Copyright 2026 by the author(s).

nou et al., 2024; Zhao et al., 2026), which formalizes an infinite-order recurrence.

In this work, we consider a unifying perspective on token mixing, showing that every causal linear mixer can be decomposed into (i) direct one-step input influence and (ii) recurrent propagation through past outputs. This structured recurrence view covers attention, SSMs, and linear attention, while exposing how design parameters control complexity, cache size, and long-range capacity.

Our contributions are the following:

- We formalize a general framework for causal linear token mixing that captures attention, SSMs, and their hybrids as special cases.
- We provide theoretical insights into the trade-offs between computational complexity and expressive power.
- We construct token mixers spanning a controlled range of complexities, from $\mathcal{O}(n)$ and $\mathcal{O}(n \log n)$ up to $\mathcal{O}(n^{3/2})$ and $\mathcal{O}(n^2)$.
- We empirically validate these designs on synthetic benchmarks and language modeling pre-training tasks.

Taken together, our results offer a principled lens for analyzing and designing efficient, expressive token mixers, providing conceptual clarity across diverse architectures.

## 2. Related works

**Attention and efficient variants.** Transformers popularized global attention, but its quadratic complexity has motivated numerous efficiency improvements. One line of work retains exact softmax attention while optimizing CUDA kernels (e.g., FlashAttention (Dao et al., 2022; Dao, 2024)). Another line alters the operator itself: sparse and local attention restrict interactions while preserving long-range connectivity (Child et al., 2019; Beltagy et al., 2020; Zaheer et al., 2020), while low-rank or kernelized linear attention formulations reduce complexity via feature maps or projections (Katharopoulos et al., 2020; Choromanski et al., 2021; Wang et al., 2020; Xiong et al., 2021).

**State Space Models (SSMs).** An alternative approach frames token mixing as a linear dynamical system. HiPPO-based methods project sequences onto orthogonal polynomial bases to retain long-range history (Gu et al., 2020), while S4 and successors use diagonal-structured operators that can be implemented efficiently, either as convolutions or through parallel scan algorithms (Gu et al., 2022c; Smith et al., 2023). Adaptive variants, such as Mamba (Gu & Dao, 2024), introduce input-dependent gating to handle more complex sequence patterns, and Mamba-2 (Dao & Gu, 2024) streamlines the recurrence while providing theoretical connections to linear attention: its structured recurrence is equivalent to a 1-semiseparable transformation matrix,

linking SSMs with masked linear attention and other gated linear attention variants.

**Theoretical limitations of SSMs and Transformers.** SSMs provide efficient linear recurrences, but their memory of past inputs decays exponentially with distance (Wang et al., 2025b), limiting long-range dependencies. Transformers, in contrast, can attend globally but lack recurrence, making tasks like entity tracking or copying challenging (Jelassi et al., 2024; Fagnou et al., 2024), and hindering generalization to longer sequences than seen in training (Beck et al., 2024).

**Hybrids.** Many recent models combine mixers to exploit complementary strengths. For example, Griffin interleaves gated linear recurrence with local attention (De et al., 2024), and B'MOJO integrates SSMs, local, and sparse attention in a single layer (Zancato et al., 2024). Larger models, such as Nemotron-H and Gemma 3, mix SSM and attention layers across the network to balance efficiency and expressivity (NVIDIA et al., 2025; Team et al., 2025a). Other works explore combinations of attention and SSM-like operators (Waleffe et al., 2024; Wang et al., 2025a; Arora et al., 2024b; Thomas et al., 2025). These hybrid designs motivate frameworks that can describe multiple token mixing mechanisms within a single mathematical form.

**Higher-order recurrence.** Beyond first-order recurrences, a few works explore multi-step or infinite-order dependencies. Higher-order RNNs were studied classically (Hush et al., 1991; Soltani & Jiang, 2017). More recently, log-linear attention exhibits a logarithmic-order recurrence (Guo et al., 2026), while ChaCAL implements an infinite-order recurrence with explicit causal structure (Fagnou et al., 2024). Block-Chacal (Zhao et al., 2026) improves the complexity of Chacal by decoupling local and long-term dependencies. In another line of work, Chimera (Lahoti et al., 2025) generalizes SSMs to graphs, which can indirectly be used to compute higher-order recurrences since a sequence can be represented as a graph. These examples motivate our generalization of recurrence patterns beyond the standard first-order view.

It was shown by Chen et al. (2025) that models require a memory capacity growing faster than $L^{\beta}$, where $L$ is the sequence length and $0 < \beta \leq 1$ is a task-dependent scalar. This implies that fixed-order recurrences will always struggle for long sequences, and motivates our approach where the order of the recurrence increases with $L$.

## 3. Framework

We consider causal token mixing as a linear operator that cleanly separates (i) direct, single-hop contributions from inputs and (ii) recursive, multi-hop contributions propagated through past outputs.

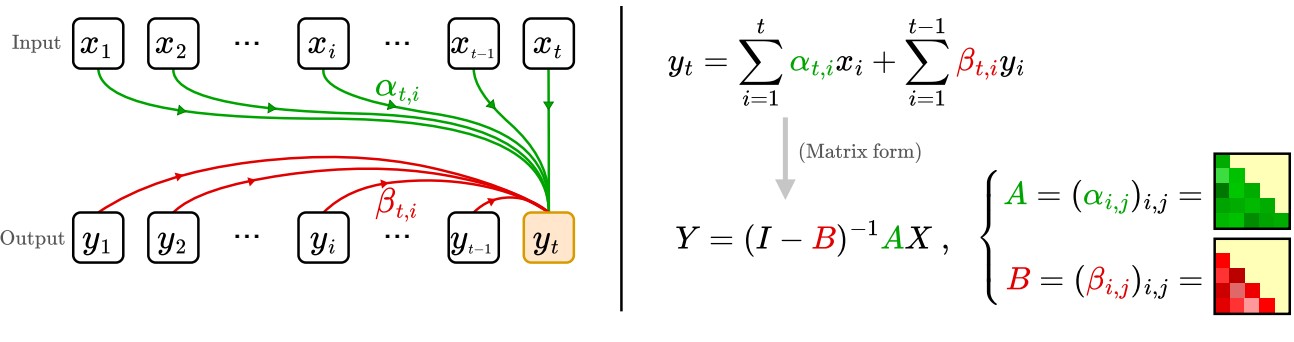

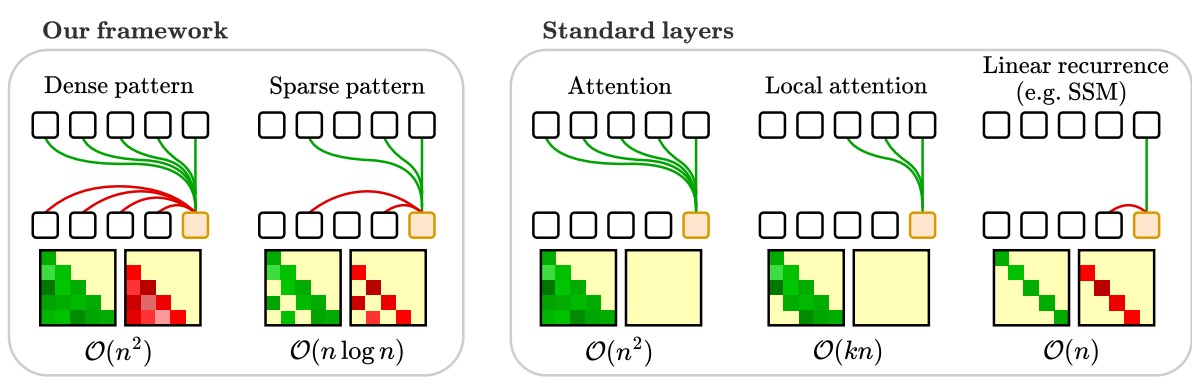

Figure 1. (Top) Our general formulation of a linear token mixing layer, which combines attention coefficients (green) and recurrence coefficients (red). The output can be expressed simply using matrix notations. (Bottom) Examples of how different sparsity patterns of $A$ and $B$ produce standard layers (attention, linear recurrence, etc.) but also enable new behaviors.

### 3.1. Recurrent and matrix forms

**Definition 3.1** (Generalized linear recurrence layer). Given a sequence of $n$ input vectors $x_1, \ldots, x_n \in \mathbb{R}^d$, we define a generalized linear recurrence as a layer which outputs the vectors $y_1, \ldots, y_n \in \mathbb{R}^d$, such that:

$$y_i = \underbrace{\sum_{j=1}^{i} \alpha_{i,j} \, x_j}_{\text{past inputs (direct mixing)}} + \underbrace{\sum_{j=1}^{i-1} \beta_{i,j} \, y_j}_{\text{past outputs (recurrent mixing)}} \qquad (1)$$

where the coefficients $\alpha_{i,j}$ and $\beta_{i,j}$ are arbitrary functions of the inputs (e.g. attention weights, SSM gating, etc – see Appendices A and B for a discussion of possible choices).

That is, the output $y_i$ is a linear combination of the past inputs $x_1, \ldots, x_i$, and the past outputs $y_1, \ldots, y_{i-1}$. A visualization of this layer is provided in Figure 1 (top).

This recursive equation is useful for computing the outputs one at a time, at inference. However, the following matrix form has more compact notations and enables the use of more efficient parallel solvers.

**Proposition 3.2** (Matrix form). *The output of a generalized linear recurrence layer can be equivalently expressed in matrix form:*

$$Y = (I - B)^{-1} A X \qquad (2)$$

*where $X = (x_1 \ldots x_n)^\top$, $Y = (y_1 \ldots y_n)^\top$, $A = (\alpha_{i,j})_{i,j} \in \mathbb{R}^{n \times n}$ is lower triangular, and $B = (\beta_{i,j})_{i,j} \in \mathbb{R}^{n \times n}$ is strictly lower triangular. This guarantees that $I - B$ is always invertible.*

*Proof.* Equation 2 follows naturally from writing Equation 1 in the matrix form as $Y = AX + BY$. $\qquad \square$

Note that this general formulation of linear recurrence is common in control theory (Anderson et al., 2019; Sieber et al., 2022) and other fields, and that the connection has been made in previous work for linear attention variants and SSMs (Gu et al., 2022c; Sieber et al., 2024; Dao & Gu, 2024; Fagnou et al., 2024; Team et al., 2025b; Lahoti et al., 2025). We simplify the framework and extend it to include softmax-based attention, in a way that directly translates into practical use. In particular, it highlights the importance of matrices $A$ and $B$, and how their structure (e.g. sparsity pattern) controls expressivity and computational cost.

### 3.2. Attention and linear recurrences as special cases

By choosing specific structures for the matrices $A$ and $B$, a generalized linear recurrence layer can behave like diverse standard token mixing layers:

- **Attention**: When $B = 0$, the layer output becomes $Y = AX$ which is exactly the attention layer where

$A$ is the attention matrix and $X$ the values. Further structure on $A$ (e.g. banded matrix) produces sparse attention variants (e.g. local attention).

- **Linear recurrence**: If $B$ is subdiagonal and $A$ diagonal, the recurrence is $y_t = \alpha_{t,t} x_t + \beta_{t,t} y_{t-1}$, which is a gated linear recurrence.
- **Diagonal SSMs**: These are in fact a special case of linear recurrence, with specific parameterization, and state expansion.

These examples are illustrated in Figure 1 (bottom). We provide more detailed explanations regarding how these layers fit in our framework in Appendix A.

## 4. Pattern design

An advantage of representing token mixing with Equation 2 is that we can enforce structure on $A$ and $B$ while still maintaining good expressivity. Indeed even if $B$ is very sparse, the inverse $(I - B)^{-1}$ will be a dense lower-triangular matrix which may model complex behaviors. In this section we explore how the structure of $A$ and $B$ influences time and memory complexities, as well as measures of expressivity. All proofs can be found in Appendix D.

### 4.1. Translation-invariant patterns

We start by investigating the class of attention patterns that are invariant under translation. They come as a simple choice, and allow us to provide a theoretical analysis of their expressivity. Here and in the rest of the paper, we consider the case where $A$ and $B$ share the same pattern.

#### 4.1.1. DEFINITIONS

Consider the equation 1 with $1 \le j \le i \le n$, the token $i$ depends on all the tokens of index $j$ such that $\alpha_{i,j} \ne 0$.

**Definition 4.1** (Translation-invariant pattern). Let $f : \mathbb{N}_{\ge 0} \to \mathbb{N}_{>0}$ be a strictly increasing function. We say a generalized linear recurrence layer follows the translation-invariant pattern induced by $f$, if:

$$\forall 1 \le j \le i \le n, \quad \alpha_{i,j} \ne 0 \text{ or } \beta_{i,j} \ne 0 \\ \Leftrightarrow \quad \exists k \in \mathbb{N} \text{ s.t. } j = i - f(k) \tag{3}$$

In other words, a token $i$ can only attend tokens $j$ that are at a distance $f(k)$ in the past, for the different and admissible values of $k$. For example, if $f(k) = 2^k$, a token at position $i$ will only be allowed to attend the positions $i - 1, i - 2, i - 4, i - 8 \ldots i - 2^{\lfloor \log_2 i \rfloor}$. The upper part of Figure 2 shows how the matrices $A$ and $B$ look like. We see that this exponential $f$ leads to a logarithmic number of past indices, which we generalize in the following proposition.

**Proposition 4.2** (Time complexity). *The token mixing layer with pattern induced by $f$ has a time complexity in $\mathcal{O}(g(i))$*

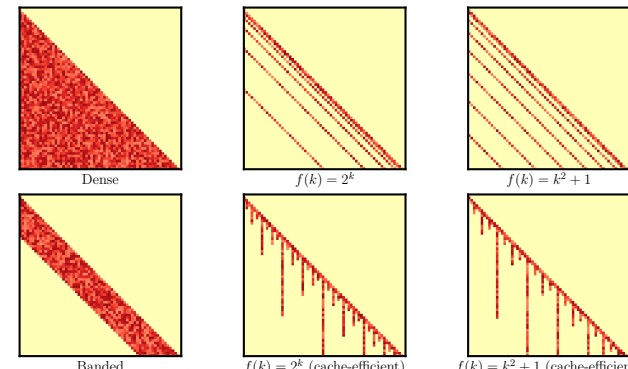

*Figure 2.* Visualization of different sparsity patterns for the matrices $A$ and $B$. Non-zero entries are colored in red. The time and space complexities of the layer depend on the properties of the sparsity pattern.

*for decoding the $i$-th token, where:* $g(i) = \max\{k \in \mathbb{N} \text{ s.t. } f(k) < i\}$.

*If $f$ can be extended to an invertible real function $\mathbb{R} \to [1, \infty)$ then the complexity is in $\mathcal{O}(f^{-1}(i))$. We will assume this is the case in the rest of the paper for simplicity.*

For example we may choose $f$ to be linear, quadratic or exponential, which will imply a time complexity respectively linear, square-root, and logarithmic.

Finally, we introduce the communication graph $\mathcal{G}$, which we will use in the next sections to analyze information propagation across the recurrence, and measure expressivity.

**Definition 4.3** (Communication graph). We consider the communication graph $\mathcal{G} = (V, E)$, where the vertices $V = \{1, 2, \ldots, n\}$ represent the token positions, and there is an edge from $j$ to $i$ iff $i - j = f(k)$ for some $k \in \mathbb{N}$.

$\mathcal{G}$ is a directed acyclic graph (DAG) and intuitively, this graph represents how information can travel from a position to another, when computing the output of the layer. Two positions are connected if the pattern induced by $f$ allows this connection, which means a nonzero value in the $A$ and $B$ matrices. For a given node at position $i$, the set of incoming edges tells you the set of tokens that attend it.

#### 4.1.2. SHORTEST INFORMATION PATH

One metric for measuring expressivity in such models is the shortest path that information can follow from a token $j$ to a token $i > j$ in the communication graph $\mathcal{G}$. Indeed, while in an attention layer all tokens are directly connected (distance of 1), recurrent models struggle at capturing long-range dependencies (Wang et al., 2025b), since information has to be stored in memory for a long period of time. The longer the information path is, the harder it is to learn, especially because of vanishing gradient effects.

**Proposition 4.4** (Shortest path). *Given two positions $j < i$*

*Table 1.* Different structures of the token mixing layers provide different trade-offs between computational cost and expressivity. We measure expressivity both using theoretical tools (section 4.1) and synthetic benchmarks (section 5.2). We compare standard attention, and its local version, to diverse structures: a diagonal SSM (first-order recurrence), a $k$-th order recurrence, and a dense recurrence (infinite order), followed by the patterns indiced by an exponential and a quadratic functions, as well as their cache-efficient version.

| | Computational cost | | Expressivity | | Synthetic tasks (%) | | |
| **Structure** | **Time per token** | **Cache size** | **Shortest path between tokens** | **Congestion** | **Copy** | **Associative recall** | **Multi-hop recall** |
|---|---|---|---|---|---|---|---|
| Attention | $\mathcal{O}(n)$ | $\mathcal{O}(n)$ | 1 | 1 | **100.00** | **100.00** | 39.21 |
| Local attention | $\mathcal{O}(k)$ | $\mathcal{O}(k)$ | $\infty$ | 1 | 23.75 | 26.20 | 23.59 |
| Diagonal SSM | $\mathcal{O}(1)$ | $\mathcal{O}(1)$ | $n$ | $n$ | 42.98 | 32.53 | 27.17 |
| $k$-th order recurrence | $\mathcal{O}(k)$ | $\mathcal{O}(k)$ | $\frac{n}{k}$ | $\frac{n}{k}$ | 74.66 | 41.12 | 39.08 |
| Dense recurrence | $\mathcal{O}(n)$ | $\mathcal{O}(n)$ | 1 | 1 | **100.00** | **99.99** | **99.80** |
| $f(k) = 2^k$ | $\mathcal{O}(\log_2 n)$ | $\mathcal{O}(n)$ | $\leq \log_2 n$ | $\leq \log_2 n$ | 92.63 | 49.03 | 34.85 |
| + cache-efficient | $\mathcal{O}(\log_2 n)$ | $\mathcal{O}(\log_2 n)$ | | | 75.47 | 52.59 | 38.63 |
| $f(k) = k^2 + 1$ | $\mathcal{O}(\sqrt{n})$ | $\mathcal{O}(n)$ | $\leq 4$ | $\leq 4$ | **99.66** | 53.61 | 35.68 |
| + cache-efficient | $\mathcal{O}(\sqrt{n})$ | $\mathcal{O}(\sqrt{n})$ | | | 91.59 | 54.56 | 38.02 |

in the communication graph $\mathcal{G}$, the length of the shortest path from $j$ to $i$ is:

$$d(i,j) = \min \left\{ d \in \mathbb{N} \ s.t. \ \exists\, a \in \mathbb{N}^d, \sum_{k=1}^{d} f(a_k) = i - j \right\} \tag{4}$$

*That is, the length depends on how many values of $f$ are needed to decompose the integer $i - j$.*

**Corollary 4.5.** *While Equation 4 is a complex problem to solve, simple choices for $f$ lead to closed-form solutions:*

- *If $f(k) = 2^k$, then $d(i,j)$ is the number of ones in the binary representation of $i - j$. This gives the bound $d(i,j) \leq \log_2(i - j)$.*

- *If $f(k) = k^2 + 1$, then by Lagrange's four-square theorem, we find that $d(i,j) \leq 4$.*

### 4.1.3. CONGESTION

A major problem of standard recurrent models is that all past information is compressed into a single vector, which makes it impossible to recall large pieces of information (Jelassi et al., 2024; Chen et al., 2025). By introducing additional connections to older hidden states, we aim at alleviating this bottleneck.

We formalize this via *graph congestion*, in a setup similar to Jelassi et al. (2024). The model is tasked with copying a sequence of length $n$ from input positions $1, \ldots, n$ to output positions $n+1, \ldots, 2n$. In order to succeed, the model must find a way to route each input token $i$ to their respective output position $i + n$.

Consider a single token-mixing layer following a time-invariant pattern induced by $f$, and $\mathcal{G}$ the communication graph on this sequence of length $2n$.

**Definition 4.6** (Congestion). Let a candidate routing $\mathcal{P}$ be a set of $n$ paths in $\mathcal{G}$, where the $i$-th path connects input node $i$ to output node $i + n$. We define the congestion of this routing as the maximum number of paths passing through any single node:

$$C(\mathcal{G}, \mathcal{P}) := \max_{1 \leq i \leq 2n} \#\{\, p \in \mathcal{P} \mid i \in p \,\}. \tag{5}$$

The congestion of the layer is then the congestion of the best routing:

$$C(\mathcal{G}) := \min_{\mathcal{P}} C(\mathcal{G}, \mathcal{P}) \tag{6}$$

Intuitively, $C(\mathcal{G})$ measures the largest number of information paths that must pass through a single node. Standard recurrent models induce high congestion, since all paths must pass through the same positions, whereas higher-order recurrences can reduce $C(\mathcal{G})$ by distributing information across multiple states.

**Proposition 4.7** (Lower bound on congestion). *If we know that the shortest path between token $i$ and $i + n$ is at least $d$ long, for all $1 \leq i \leq n$, then we get:*

$$C(\mathcal{G}) \geq \frac{d+1}{2} \tag{7}$$

**Proposition 4.8** (Upper bound on congestion). *If the pattern is translation-invariant, and we know that the shortest path between token $i$ and $i + n$ is at most $D$ long, for all $1 \leq i \leq n$, then we get:*

$$C(\mathcal{G}) \leq D \tag{8}$$

**Corollary 4.9.** *Combining Corollary 4.5 with Proposition 4.8, we get that:*

- *If $f(k) = 2^k$, then $C(\mathcal{G}) \leq \log_2(n)$.*

- *If $f(k) = k^2 + 1$, then $C(\mathcal{G}) \leq 4$.*

Together, Propositions 4.7 and 4.8 suggest a direct link between shortest information path and congestion.

## 4.2. Cache-efficient patterns

Translation-invariant patterns achieve sublinear decoding complexity, but their cache size remains in $\mathcal{O}(n)$. Indeed, a token at position $j$ may still be attended arbitrarily far in the future, so all past keys and values must remain in memory.

We construct a cache-efficient variant by enforcing the following constraint: when decoding token $i$, attention is only allowed toward positions that were already used when decoding token $i - 1$. As a result, the active cache evolves recursively and its size remains sublinear.

Let $S_i \subseteq \{1, \ldots, i\}$ denote the active cache when decoding token $i$, i.e. the set of past positions that may be attended by token $i$. Starting from the translation-invariant offsets $f(k)$, each target position $i - f(k)$ is rounded upward to the closest position already present in the previous cache.

**Definition 4.10** (Cache-efficient pattern). First define the operator:
$$\pi_A(t) = \min([t, \infty) \cap A), \tag{9}$$

which rounds index $t$ up to the nearest element of a set $A$.

We define $S_1 = \{1\}$ and, for $i > 1$:
$$S_i = \left\{ \pi_{S_{i-1} \cup \{i\}}(i - f(k)) \mid k \in \mathbb{N}, f(k) < i \right\}. \tag{10}$$

The attention coefficients then satisfy:
$$\alpha_{i,j} \neq 0 \quad \text{or} \quad \beta_{i,j} \neq 0 \implies j \in S_i. \tag{11}$$

Equivalently, each translation-invariant offset $i - f(k)$ is rounded upward to the nearest position already present in the cache.

Figure 2 visualizes the resulting attention patterns. Unlike the original translation-invariant construction, the cache evolves recursively and only contains positions reused by future decoding steps.

**Proposition 4.11** (Time complexity and cache size). *For decoding the $n$-th token, the cache-efficient pattern has both decoding complexity and cache size in $\mathcal{O}(f^{-1}(n))$.*

Surprisingly, the recursive construction admits a simple closed-form expression revealing a periodic structure in the cache evolution.

**Proposition 4.12.** *The recursive definition 10 admits the following closed-form expression:*
$$S_i = \left\{ a_k \left\lceil \frac{i - f(k)}{a_k} \right\rceil \mid k \in \mathbb{N}, f(k) < i \right\} \tag{12}$$

*where $a_0 = 1$ and $a_{k+1} = a_k \left\lceil \frac{f(k+1) - f(k)}{a_k} \right\rceil$.*

*In particular, the cache position associated with scale $f(k)$ evolves periodically with period $a_k$.*

Equation 12 shows that the cache positions are quantized onto lattices of step size $a_k$. This has important implications when considering efficient implementations, which could leverage this structure – see discussions in Appendix B.4.

## 4.3. On the role of dimension

Our framework – and the theoretical insights we provide – focus on the temporal dimension and how information is propagated along the sequence. The actual amount of information that can be stored and propagated by the model (i.e. the actual "memory capacity") depends linearly on the channel dimension, which can thus be leveraged to increase the capacity of the model and compensate for the higher congestion. In practice, many models use much larger channel dimensions (e.g. SSMs using state-expansion) to be competitive with attention-based models while remaining linear in the sequence length.

# 5. Experiments

We validate our claims with two sets of experiments. First, we use synthetic tasks to isolate and probe specific capabilities of token-mixing layers under controlled conditions. These tasks are designed to stress the theoretical knobs introduced in our framework (path length, congestion, cache structure). Second, we assess end-to-end performance on real-world data by training language models on OpenWebText. Together, the results test whether the theoretical predictions survive contact with training dynamics and natural language statistics.

## 5.1. Models

We base our architecture on the standard transformer, and in particular on GPT-2 (Radford et al., 2019), with the exception that we use RoPE for positional embeddings (Su et al., 2024). The attention layer is swapped for one of the token-mixing layers studied in this paper, all implemented within the same backbone (same depth, dimension, normalization, MLP blocks) so that comparisons isolate the effect of token mixing. For reference, we include full attention and local attention with window size $w$=8 as baselines.

Within our framework, we compare several $(A, B)$ structures (shared sparsity for $A$ and $B$ unless otherwise noted): *dense* (lower-triangular) $A$ with strictly lower-triangular $B$ (full resolvent mixer), *banded* with bandwidth $w$=8, and two translation-invariant families controlled by stride functions $f$: exponential $f(k)$=$2^k$ and quadratic $f(k)$=$k^2$+1. When relevant, we also evaluate cache-efficient variants (Section 4.2), which sparsify the working set while preserv-

ing the induced access pattern. Practical aspects (parameterization and normalization of $A$ and $B$, conditioning of $(I-B)$, and implementation details) are discussed in Appendix B.

## 5.2. Synthetic tasks

### 5.2.1. SETUP

We consider three classical sequence problems adapted from prior work, chosen to stress different aspects of path geometry and memory pressure:

1. **Copy**: The model must copy an input sequence of size $L$ (Arjovsky et al., 2016; Jelassi et al., 2024). This task measures the ability of the model to memorize the sequence, and directly measures the congestion in the token mixing layers.
2. **Associative recall**: A similar yet more challenging task, where the model is given a series of key-value pairs that it must memorize. Then, when queried the keys, it must output the corresponding values. This measures whether the mixer can maintain a structured, addressable memory and is often used to benchmark SSM-like models (Arora et al., 2024a; Dao & Gu, 2024).
3. **Multi-hop recall**: Inspired from Fagnou et al. (2024), this task requires the model to solve a chain of associative recalls, which is particularly difficult for non-recurrent models. We modify the associative recall task by replacing some values by past keys, that the model must then recursively lookup. This measures the state-tracking ability of the models, which is required in various downstream tasks (Mavi et al., 2024).

To avoid overfitting to a single length scale, we randomize sequence lengths per batch. All models consist of two Transformer blocks; the first serves to preprocess the token stream into a representation amenable to the mixer, and the second performs the task-specific transport. Training, optimization, and sampling protocols are kept identical across models. Full hyperparameters and setup details are given in Appendix C.

### 5.2.2. RESULTS

We report the results for the synthetic tasks in Table 1.

Only the most general formulation with $A$ and $B$ dense is able to perfectly solve all tasks. While standard attention works as well for the copy and associative recall tasks, it struggles on multi-hop recall, which is expected (Fagnou et al., 2024). Local attention performs poorly across all settings.

The banded structure performs relatively well but falls behind patterns that involve more long-distance connections.

The pattern induced by $f(k) = k^2 + 1$ seems better than $f(k) = 2^k$, although not by a big margin. $f(k) = k^2 + 1$ is the only one able to reach a near perfect score on the copy task, which is expected since we showed its congestion is at most 4. Its performance is however not as great for the recall tasks, which can be due to the difficulty of learning the optimal paths (Wen et al., 2025; Kim & Suzuki, 2025).

The results appear especially encouraging for the cache-efficient variants. They even sometimes outperform their original counterparts, which is surprising. Still, this suggests that sparsifying the cache does not necessarily reduce the expressivity of the layer.

## 5.3. Language modeling

In this section we evaluate the models on a language modeling task using the OpenWebText dataset (Gokaslan & Cohen, 2019). The goal is to confirm that the theoretical insights, and the results on synthetic tasks, can transfer to real natural language. We train three models with respectively 125M, 355M and 775M parameters, while replacing attention with various token mixing layers. Context size is 1024 for the smaller model, to 2048 for the larger ones. The full experimental setup is detailed in Appendix C.

**Results Analysis** Here, we analyze the results presented in Figure 3, first by comparing results inside each class and then comparing every trained models' performance.

**Comparison *inside* each class.**

1. $O(n)$ *time.* Within the full-complexity regime, the layer with dense lower-triangular $A$ and strictly lower-triangular $B$ consistently sits slightly below or matches standard full-attention. Intuitively, $B$ accumulates causal summaries; the resolvent $(I - B)^{-1}$ expands them into a dense, geometry-aware receptive field; $A$ then reprojects, yielding more effective long-range mixing than dot-product attention for the same complexity.
2. *Sublinear variants* $(O(\sqrt{n}) - O(\log n))$. First, we observe that the $O(\sqrt{n})$ model gives the better results in this bracket, corroborating with its denser $B$ matrix. Among structured sublinear designs, we observe a tight low-perplexity frontier, with two distinct behaviors. Cache-efficient variants always perform slightly worse than their time-invariant counterparts. While it is expected to have a gap, it is still encouraging to see that this gap remains small, suggesting that the cache-efficient transformation does not worsen the expressivity significantly, while greatly improving memory-efficiency. Lastly, in this setting, it is interesting to note that the standard attention-based model (local attention) is largely beaten by its recurrent counterpart in $\mathcal{O}(k)$.
3. *Recurrent* $O(1)$. Constant-time models cluster at

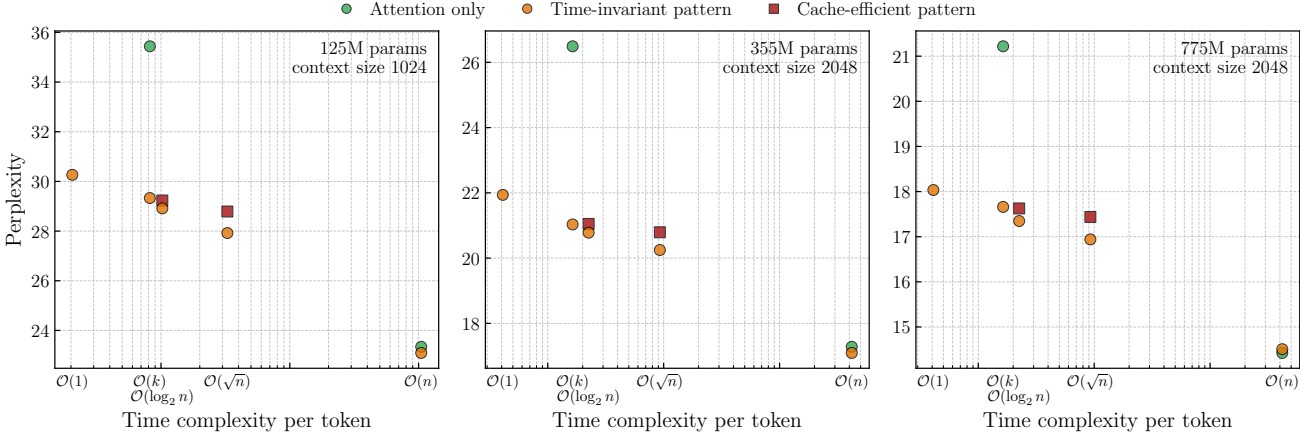

*Figure 3.* OpenWebText perplexity (lower is better) versus per-token time complexity for respectively 125M, 355M and 775M parameters models. Points correspond to attention-based, recurrent, and cache-efficient variants; the $x$-axis annotates canonical regimes $O(1)$, $O(k)$, $O(\log_2 n)$, $O(\sqrt{n})$, and $O(n)$.

higher perplexities with comparatively small spread. This aligns with the congestion view: compressing the entire past into a single state under-exploits long-range dependencies under the given budget. This proposition still beats the local attention method.

**Comparison *across* classes: the Pareto frontier.** Taken together, the points trace a clear speed-accuracy Pareto curve. Large gains occur when moving off $O(1)$ to sublinear access; by $O(\log n)$, diminishing returns appear, with several cache-efficient models matching the attention band – thus recovering most of attention's accuracy at substantially lower theoretical cost. If $O(n)$ is affordable, the resolvent mixer "wins the bracket," surpassing standard attention without a significant parameter increase; if not, well-designed $O(\sqrt{n})$ or $O(\log n)$ caches provide competitive perplexity at a fraction of per-token complexity.

## 6. Discussion

The factorization $y = (I - B)^{-1}Ax$ provides a framework-level view of causal token mixing, where $A$ captures the direct input–output interactions within a single step and $B$ governs the recurrent propagation of information across steps. Varying their sparsity patterns and structure spans a range of architectures often treated as distinct, while keeping strict causality and triangular solves explicit. Translation-invariant patterns parameterized by a strictly increasing function $f$ make complexity and geometry analyzable: the decoding cost scales as $O(f^{-1}(n))$, shortest-path lengths are tied to integer decompositions of $i-j$, and congestion bounds characterize expressivity bottlenecks. Choices such as $f(k) = 2^k$ (logarithmic time, logarithmic depth) or $f(k) = k^2 + 1$ (square-root time, constant depth) illustrate that "global vs. local" is not a binary; it is a tunable Pareto surface, with hop depth, connectivity, and cache size

jointly controlled by $(A, B)$ and the pattern $f$.

Empirically, this perspective helps organize results across complexity regimes. Cache-efficient variants – restricting attention to indices already present in the previous step's cache – reach $O(f^{-1}(n))$ complexity for both time and memory, and often match, or only slightly underperform, their non-cache counterparts. This suggests that structured choices for $B$ act as a useful inductive bias rather than a limitation. Aggregating models reveals a clear speed–accuracy frontier: moving from $O(1)$ to sublinear access yields substantial gains, and by $O(\sqrt{n})$ the proposed designs recover a significant fraction of the performance gap to full attention. When $O(n)$ complexity is acceptable, the fully general resolvent mixer $(I - B)^{-1}A$, with dense lower-triangular $A$ and strictly lower-triangular $B$, outperforms standard full attention at comparable per-token cost with only a marginal increase in parameters. In practice, this leads to a simple design procedure: (i) choose a target complexity class based on system constraints, (ii) select $f$ to control hop geometry and congestion, (iii) enforce cache efficiency to improve memory locality, and (iv) adjust the parameterization of $A$ to balance expressivity and stability.

Limitations and system implications follow directly. Our experiments involve relatively small models and unoptimized kernels; realizing end-to-end speedups requires specialized block-triangular forward-substitution kernels with regular memory access that exploit the periodic structure of cache-efficient patterns. While we discuss several practical considerations in Appendix B, there remain challenges to solve in order to make such token mixing layers scale efficiently and compete against other models. Stability and optimization might depend on parameterizations that preserve simple invariants; alternative parameterizations could change training dynamics. Scaling studies at longer contexts

and larger model sizes, heterogeneous layers mixing different $(A, B)$ structures, principled initializations for $B$ (for example inspired by the works on structured initializations for SSM (Gu et al., 2022b)), and parameter-efficient factorizations for $A$ and $B$ (low-rank, semiseparable, or Toeplitz) are natural next steps to test whether the observed Pareto frontier persists at LLM scale.

## 7. Conclusion

This work reframes causal token mixing as a small set of explicit, controllable design choices, turning "which architecture?" into "which geometry and budget?". Rather than reiterating the factorization or translation-invariant analysis, we emphasize the shift in practice: pick a target complexity class, shape hop geometry to manage path lengths and congestion, and deploy cache-aware implementations that honor those choices. Our experiments anchor the theory by showing that sublinear access captures most of the accuracy of dense mixing at far lower budget, and that within the $\mathcal{O}(n)$ bracket a resolvent-style mixer can outperform standard attention with only marginal parameter overhead.

Looking forward, two tracks are most promising. On the *systems* side, specialized triangular-solve kernels and cache layouts are the lever to translate asymptotic gains into latency/throughput improvements at long context. On the *modeling* side, relaxing strict translation invariance toward learned, data-dependent patterns while preserving analyzable path and cost guarantees could broaden the design space without losing clarity. Our aim is not to crown a single mixer, but to provide a principled toolkit through which future architectures can be designed, analyzed, and engineered coherently.

## Acknowledgements

This work was granted access to the HPC resources of IDRIS under the allocations AD011015154R2 and A0191016927 made by GENCI. It has received support from the French government, managed by the National Research Agency, under the France 2030 program with the reference "PR[AI]RIE-PSAI" (ANR-23-IACL-0008) and "PEPR-SHARP" (ANR-23-PEIA-0008). This work was also partially supported by JST PRESTO JP-MJPR23P5, JST CREST JPMJCR21M2, JST NEXUS JP-MJNX25C4.

## Impact Statement

This paper presents work whose goal is to advance the field of Machine Learning. There are many potential societal consequences of our work, none of which we feel must be specifically highlighted here.

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

# A. Examples encompassed by the framework

The framework aims at generalizing standard token mixing layers into a unified view, while opening the way to a new class of layers. The following section describes in more details how standard layers fit in the framework as special cases of the structure of $A$ and $B$ from Equation 2, which control the token interactions.

## A.1. Attention

Given input vectors $U = (u_1 \ldots u_n)^\top \in \mathbb{R}^{n \times d}$, the (self) attention layer (Vaswani et al., 2017) can be obtained using our framework by picking:

$$
\begin{cases}
A = \text{softmax}\left(\frac{1}{\sqrt{d_{\text{head}}}} U W^q W^{k\top} U^\top + M\right) \\
B = 0 \\
X = U W^v
\end{cases}
\tag{13}
$$

where $W^q$, $W^k$, $W^v$ are respectively the query, key and value weight matrices, and $M$ is a mask matrix that enforces causality. Note that the input of the attention $U$ is linearly transformed using $W^v$ to obtain the values, which correspond the input $X$ in our framework.

## A.2. Sparse / local attention

Attention variants that utilize special attention patterns also fit in the framework. In Equation 13, the mask matrix $M$ may be used to mask out arbitrary entries of the attention matrix, thus enforcing sparse structure. This is the case in local attention, dilated local attention (Beltagy et al., 2020), and other sparse attention mechanisms (Zaheer et al., 2020; Roy et al., 2021).

## A.3. Linear recurrence

Despite the success of RNNs, and in particular gated RNNs such as GRUs and LSTMs, their sequentiallity limits their scale. Several works (Bradbury et al., 2017; Stanić et al., 2023; Feng et al., 2024; De et al., 2024) have proposed simplifications to the gating mechanism, removing the dependency of the gating coefficients to the previous hidden state. This modification allows the use of efficient parallel scan algorithms, unlocking larger model scales. Formally, these models all follow a similar equation for computing the hidden states $h_t \in \mathbb{R}^d$:

$$
h_t = r_t \odot h_{t-1} + i_t \odot x_t
\tag{14}
$$

where $r_t \in \mathbb{R}^d$ is the *reset* gate, $i_t \in \mathbb{R}^d$ is the *input* gate, and both are functions of the current input $x_t \in \mathbb{R}^d$.

To fit these in our framework, we need to consider this equation elementwise (or equivalently setting $d = 1$ and then stacking $d$ independent heads). Equation 14 becomes, in our framework's notations:

$$
y_t = \beta_{t,t-1} y_{t-1} + \alpha_{t,t} x_t
\tag{15}
$$

which means, in terms of matrices $A$ and $B$:

$$
\begin{cases}
A = \begin{pmatrix} i_1 & & 0 \\ & \ddots & \\ 0 & & i_n \end{pmatrix} \\
B = \begin{pmatrix} 0 & & & 0 \\ r_2 & & & \\ & \ddots & & \\ 0 & & r_n & 0 \end{pmatrix}
\end{cases}
\tag{16}
$$

### A.4. State-space models

State-space models are ruled by the following equations, which represent the discretization of a particular 1-dimensional continuous system:

$$\begin{cases} h_t = \mathbf{A}_t h_{t-1} + \mathbf{B}_t u_t \\ y_t = \mathbf{C}_t h_t \end{cases} \tag{17}$$

where the matrices $\mathbf{A}_t \in \mathbb{R}^{N \times N}$, $\mathbf{B}_t \in \mathbb{R}^{N \times 1}$ and $\mathbf{C}_t \in \mathbb{R}^{1 \times N}$ can be freely parameterized (Gu et al., 2022b; Fu et al., 2023), and possibly be time-dependent (Gu & Dao, 2024; Dao & Gu, 2024). $N$ is the state expansion factor.

In practice, the most recent models all use a diagonal transition matrix $\mathbf{A} = \mathrm{diag}(a_1, \ldots, a_N)$ (Gu et al., 2022a), such that we can consider Equation 17 elementwise in our framework (or equivalently use $N = 1$):

$$y_t = \beta_{t,t-1} y_{t-1} + \alpha_{t,t} x_t \tag{18}$$
$$z_t = \mathbf{C}_t y_t \tag{19}$$

where the SSM output $z_t$ is a linear projection of the recurrence output $y_t$. In terms of matrices $A$ and $B$, we have:

$$\begin{cases} A = \begin{pmatrix} \mathbf{B}_1 & & 0 \\ & \ddots & \\ 0 & & \mathbf{B}_n \end{pmatrix} \\ B = \begin{pmatrix} 0 & & & 0 \\ \mathbf{A}_2 & & & \\ & \ddots & & \\ 0 & & \mathbf{A}_n & 0 \end{pmatrix} \end{cases} \tag{20}$$

Important note: our notations differ from SSMs, and in particular here the attention matrix $A$ is akin to the input projection matrix $\mathbf{B}$, while the recurrence matrix $B$ replaces the transition matrix $\mathbf{A}$.

**Mamba.**  In the specific case of Mamba (Gu & Dao, 2024), the element-wise equation for the scalar $h_i$ is:

$$h_i = \exp(\Delta_i a) h_{i-1} + \Delta_i b_i x_i \tag{21}$$

where $\Delta_i$ is a small (input-dependent) step size, $a$ is a learnable parameter, and $b_i$ is a function of the input tokens. This can be expressed in our framework by defining $A$ and $B$ as follows:

$$\begin{cases} A = \begin{pmatrix} \Delta_1 b_1 & & & 0 \\ & \Delta_2 b_2 & & \\ & & \ddots & \\ 0 & & & \Delta_n b_n \end{pmatrix} \\ B = \begin{pmatrix} 0 & & & 0 \\ \exp(\Delta_2 a) & & & \\ & \ddots & & \\ 0 & & \exp(\Delta_n a) & 0 \end{pmatrix} \end{cases} \tag{22}$$

## B. Practical considerations

### B.1. Parameterization of $A$ and $B$

Our framework does not make any assumption on how the coefficients $\alpha_{ij}$ and $\beta_{ij}$ are computed. That is, they could be constant (Katharopoulos et al., 2020), depend on the input (Dao & Gu, 2024; Yang et al., 2024) or not (Gu et al., 2022c), the distance (Sun et al., 2023), etc.

In an effort to bridge the gap between attention and SSMs, in all our experiments we choose attention-like coefficients for both $\alpha_{ij}$ and $\beta_{ij}$. That is, $A$ and $B$ are computed like two independent attention matrices, with the addition of a gating system to ensure normalization (see Section B.2). The gating and its initialization are adapted from Mamba-2 (Dao & Gu, 2024), and was observed to help in all pattern choices for $B$. However more parameter-efficient choices may be considered in future work and may improve the overall efficiency.

In Table 2 we show that in the specific case of first-order recurrence (e.g. SSMs), using the Mamba parameterization instead of ours provides a small yet noticeable gain:

*Table 2.* Comparison of first-order recurrence models using either our parameterization of $A$ and $B$, or the Mamba rule (Gu & Dao, 2024). We report the perplexity for the 125M parameters model, using the same experimental setup as the other experiments.

| Parameterization | Perplexity |
|:---:|:---:|
| Ours | 31.28 |
| Mamba | 30.64 |

## B.2. Normalization

A recurring problem in recurrent models is vanishing and exploding gradients. To prevent such phenomenon, one ought to carefully normalize the weights in $A$ and $B$. This is key for allowing the model to learn meaningful representations.

From Equation 1, we can see that if we assume that all $||x_j|| \leq C$, and $||y_j|| \leq C$ for $j < i$, we get:

$$||y_i|| \leq C \left[ \sum_{j=1}^{i} \alpha_{ij} + \sum_{j=1}^{i-1} \beta_{ij} \right] \tag{23}$$

We then only need to ensure that $\sum_{j=1}^{i} \alpha_{ij} + \sum_{j=1}^{i-1} \beta_{ij} = 1$, or in matrix notations: $(A + B)\,\mathbf{1} = \mathbf{1}$. In practice this can be done by weighting $A$ and $B$. Fagnou et al. (2024) do this using a fixed hyperparameter $\gamma$ and use $A = (1 - \gamma)\widetilde{A}$ and $B = \gamma\widetilde{B}$. We generalize this with an input-dependent gating vector $g \in [0;1]^d$ and using $A = (1 - \text{diag}(g))\widetilde{A}$ and $B = \text{diag}(g)\widetilde{B}$.

However, one could also apply a softmax normalization to the full transformation matrix $T = (I - B)^{-1}A$. As discussed in Dao & Gu (2024), this can be achieved by computing the forward pass $Tx$ without normalizing, while computing $T\,\mathbf{1}$ at the same time and using it to normalize the output. Unfortunately, while this is mathematically sound, in the general case values skyrocket and overflow, causing numerical errors. We leave solving this numerical stability problem for future work.

## B.3. Weight sharing

While $A$ and $B$ play different roles, we could still expect them to be correlated. In this section we investigate the effect of a weight sharing of the form:

$$A = BD + D' \tag{24}$$

where $D$ and $D'$ are diagonal matrices. This is similar yet more general than Fagnou et al. (2024). Note that all the previous theoretical results still stand, since there was no assumption on $A$ and $B$ other than their sparsity pattern.

It turns out that the forward equation simplifies nicely:

$$\begin{aligned} y &:= (I - B)^{-1}(BD + D')x \\ &= (I - B)^{-1}(D + D')x - Dx \end{aligned} \tag{25}$$

Since addition and multiplication of diagonal matrices is linear and negligible, the only costly operation that remains is the multiplication by $(I - B)^{-1}$, and **the multiplication by $A$ disappeared**.

Looking now at the recursive form, by introducing the variable $z_i := y_i + d_i x_i$, we obtain:

$$z_i = \sum_{j=1}^{i-1} \beta_{ij} z_i + (d_i + d_i')x_i \tag{26}$$

The recurrence got much simpler, and in particular the **cache size is reduced** since we only need to store the useful past $z_j$, instead of both the $x_j$ and $y_j$ in the general case.

### B.4. Efficient implementations

While implementing a custom CUDA kernel to perform the sparse triangular solves efficiently is out of the scope of this paper (we used the native torch function which is always quadratic) we discuss practical implementations in this section.

The causal token-mixing operators we study induce lower-triangular matrices $M \in \mathbb{R}^{n \times n}$ with sparse, repetitive structure. Forward substitution on such matrices has complexity proportional to the number of nonzeros. If each row has at most $w$ nonzeros, then solving $Mx = b$ requires $O(nw)$ operations, compared to $O(n^2)$ for dense triangular solves.

**Block forward substitution.**   Because the sparsity patterns we consider are *periodic* along the diagonal (see Proposition 4.12), the system can be naturally partitioned into blocks, where each block shares the same nonzero structure. This allows the solve to be reorganized into a sequence of block updates:

$$x^{(k)} = M_{kk}^{-1}\Big(b^{(k)} - \sum_{\ell < k} M_{k\ell} x^{(\ell)}\Big),$$

where $M_{kk}$ denotes the $k$-th diagonal block. Each block update involves small dense solves (with identical shape across blocks), which can be vectorized and batched efficiently on GPUs.

**Parallelism.**   While the numerical entries of $M$ vary, the periodicity ensures that the memory access pattern and dependency graph repeat exactly. This enables highly regular parallel implementations: a single kernel can encode the substitution pattern once, and apply it across all blocks with different coefficients. Such regularity improves cache efficiency and load balancing compared to generic sparse triangular solvers.

**Hierarchical structure.**   If the sparsity pattern itself is recursive (e.g., defined hierarchically or via dilation), then one could further apply divide-and-conquer strategies, solving the system by recursively eliminating larger and larger blocks. This approach can reduce synchronization costs and naturally exposes parallelism across scales, complementing the block forward substitution scheme.

## C. Experimental details

### C.1. Datasets

**Synthetic tasks.**   All three synthetic datasets are generated on the fly during training, such that there is no overfitting problem. We employ some form of curriculum training as in (Dao & Gu, 2024), with the training being split into 4 phases which divide the sequence length (and other task-specific parameters if suited) by respectively 8, 4, 2 and 1.

**Copy.**   The copy task is adapted from Arjovsky et al. (2016) and Jelassi et al. (2024). The model must copy sequences with length up to $L = 128$. The beginning and end of the input sequence are marked by special tokens.

**Associative recall.**   We adapt this task from Arora et al. (2024a). We use up to 64 key-value pairs, and sequences up to 256 long. We additionally randomize more the position of the keys and values to prevent any bias favoring a specific attention pattern.

**Multi-hop.**   We use the same setup as the associative recall task, but each value has a probability $p = 0.5$ to be replaced by a preceding key. Since the task is harder to learn, we also add labels to the intermediary keys to help the model learn.

**OpenWebText.**   This dataset was built to replicate the (undisclosed) training dataset of GPT-2 (Radford et al., 2019). It contains 38GB of text data from 8,013,769 documents. We use the same tokenizer as GPT-2.

### C.2. Training setup

Training is performed on single NVIDIA V100 GPUs for the synthetic tasks, and pairs of NVIDIA A100 GPUs for language modeling. We use mixed precision with FP16.

All runs use a linear warmup for the learning rate, followed by a cosine scheduler.

### C.3. Hyperparameters

We report all hyperparameters in Table 3

*Table 3.* Hyperparameters used in the different experiments.

| Name | Synthetic tasks | Language modeling | | |
|---|---|---|---|---|
| base model params | 4M | 125M | 355M | 775M |
| train steps | 20k | 20k | 25k | 25k |
| warmup steps | 2k | 1k | 2500 | 2500 |
| lr | 3e-3 | 5e-4 | 2.5e-4 | 2e-4 |
| batch size | $\geq 1024$ | 128 | 128 | 256 |
| weight decay | 0.1 | 0.1 | 0.1 | 0.1 |
| $\beta_1$ | 0.9 | 0.9 | 0.9 | 0.9 |
| $\beta_2$ | 0.98 | 0.95 | 0.95 | 0.95 |
| grad max norm | 1.0 | 1.0 | 1.0 | 1.0 |
| vocab size | 8,192 | 50,257 | 50,257 | 50,257 |
| context length | $\leq 256$ | 1024 | 2048 | 2048 |
| num layers | 2 | 12 | 24 | 36 |
| dim | 256 | 768 | 1024 | 1280 |
| ff dim | 1024 | 3072 | 4096 | 5120 |
| head dim | 64 | 64 | 64 | 64 |

## D. Proofs

### D.1. Proof of Proposition 4.2

The result is relatively straightforward. At time $n$, we attend the past indices $n - f(0), n - f(1), \ldots, n - f(i)$, as long as $n - f(i) > 0$.

The number $k_n$ of past tokens attended is:

$$k_n := \#\{i \in \mathbb{N} \mid 0 < n - f(i)\} \tag{27}$$
$$= 1 + \max\{i \in \mathbb{N} \mid 0 < n - f(i)\} \tag{28}$$
$$= 1 + \max\{i \in \mathbb{N} \mid f(i) < n\} \tag{29}$$
$$= 1 + g(n) \tag{30}$$
$$= \mathcal{O}(g(n)) \tag{31}$$

If $f$ can be extended to an invertible real function $\mathbb{R} \to [1, \infty)$, then we have by definition of $g$ that:

$$f(g(n)) \leq n \tag{32}$$
$$\iff g(n) \leq f^{-1}(n) \tag{33}$$

and hence $k_n = \mathcal{O}(f^{-1}(n))$.

### D.2. Proof of Proposition 4.4

Given two positions $i < j$, we denote $d(i, j)$ the length of the shortest path from token $i$ to token $j$.

We can consider the underlying directed acyclic graph of the pattern: each token is a node, and there is an edge $i \to j$ iff

$\exists\, k \in \mathbb{N}, f(k) = i - j.$

$$d(i,j) = \min \left\{ k \mid \exists\, v \in [1,i]^{k-1}, i \to v_1 \to \cdots \to v_{k-1} \to j \right\} \tag{34}$$

$$= \min \left\{ k \mid \exists\, v \in [1,i]^{k-1}, u \in \mathbb{N}^k, f(u_1) = v_1 - i, \ldots, f(u_k) = j - v_{k-1} \right\} \tag{35}$$

$$= \min \left\{ k \mid \exists\, u \in \mathbb{N}^k, \sum_{p=1}^{k} f(u_p) = j - i \right\} \tag{36}$$

### D.3. Proof of Corollary 4.5

While the shortest path is a nontrivial quantity in the general case, we can find exact values for simple choices for $f$:

**Exponential** $f(i) = 2^i$: A key observation is that the shortest path does not involve two edges that share the same power of 2 – otherwise they could have been replaced by a single edge uses the next power of two. Hence we are looking for a way to decompose $j - i$ into a sum of *unique* powers of two. The only solution is given by its binary representation.

**Quadratic** $f(i) = (i+1)^2$: Lagrange's four-square theorem tells us that every natural number can be written as the sum of at most 4 squares. First, we can see that when $j-i \leq 3$ this is trivially true. Consider the number $m = j-i-4$. By Lagrange's four-square theorem it can be written as $m = a^2 + b^2 + c^2 + d^2$. And hence $j - i = (a^2+1)+(b^2+1)+(c^2+1)+(d^2+1)$.

Note: while it is surprising to get a constant value, remind that exponential $f$ gives a logarithmic bound. Having a denser attention pattern should get a much better bound than logarithmic, which at our scale would appear constant.

### D.4. Proof of Proposition 4.7

Suppose we have a routing $\mathcal{P}$ containing $n$ directed paths in a graph $\mathcal{G}$, each of length at least $d$ edges. Let the graph have $2n$ nodes (for the copy task setup). By definition, a path of length $d$ edges visits $d + 1$ nodes, so the total number of node visits across all $n$ paths is at least:

$$\text{total visits} \geq n \cdot (d + 1) \tag{37}$$

Let $C(\mathcal{G}, \mathcal{P})$ denote the maximum number of paths passing through any single node. Since each of the $2n$ nodes can be traversed by at most $C(\mathcal{G}, \mathcal{P})$ paths, the total number of visits is also upper bounded by:

$$\text{total visits} \leq 2n \cdot C(\mathcal{G}, \mathcal{P}) \tag{38}$$

Combining these inequalities, we obtain:

$$n \cdot (d + 1) \leq 2n \cdot C(\mathcal{G}, \mathcal{P}) \tag{39}$$

$$\implies C(\mathcal{G}, \mathcal{P}) \geq \frac{d + 1}{2} \tag{40}$$

Since this inequality is true for any routing $\mathcal{P}$, it is also true for their minimum $C(\mathcal{G})$.

### D.5. Proof of Proposition 4.8

Consider the routing $\mathcal{P}$ where the $n$ paths have a length of $D$ edges, and where the path for input $i + 1$ is a shift of the path for input $i$ (possible since we assume the token mixing pattern is *translation-invariant*).

In this case, each node is visited by at $D$ paths simultaneously, which occurs in the overlapping region of consecutive paths. Hence, the congestion if this routing is:

$$C(\mathcal{G}, \mathcal{P}) = D \tag{41}$$

Which means the minimum routing is at most $D$.

### D.6. Proof of Proposition 4.12

If we note $p_k(i)$ the index we obtain by increasing $i - f(k)$ until reaching a cached token (with $i > f(k)$), we can write:

$$p_k(i) = \begin{cases} p_k(i-1) & \text{if } 0 < i - f(k) \le p_k(i-1) \\ p_{k-1}(i-1) & \text{else.} \end{cases} \tag{42}$$

$$= p_{k-1}(f(k)) + p_{k-1}(f(k)) \left\lfloor \frac{i - f(k) - 1}{\underbrace{p_{k-1}(f(k))}_{a_k}} \right\rfloor \tag{43}$$

$$= a_k \left( 1 + \left\lfloor \frac{i - f(k) - 1}{a_k} \right\rfloor \right) \tag{44}$$

$$= \text{"the smallest multiple of } a_k \text{ that is strictly greater than } i - f(k) - 1\text{"} \tag{45}$$

$$= \text{"the smallest multiple of } a_k \text{ that is greater or equal to } i - f(k)\text{"} \tag{46}$$

$$= a_k \left\lceil \frac{i - f(k)}{a_k} \right\rceil \tag{47}$$

We can use this equation to find a recursive relation for $p_{k-1}(f(k) - 1)$:

$$a_k := p_{k-1}(f(k)) \tag{48}$$

$$= a_{k-1} \left\lceil \frac{f(k) - f(k-1)}{a_{k-1}} \right\rceil \tag{49}$$

