# OpenReview forum: "Trading Complexity for Expressivity Through Structured Generalized Linear Token Mixing"
_ICML.cc/2026/Conference — ICML 2026 regular_

### Official Review · Reviewer_AXta · 2026-03-11

**Soundness:** 2
**Presentation:** 3
**Significance:** 2
**Originality:** 2
**Overall Recommendation:** 4
**Confidence:** 3

**Summary:**

This paper proposes a framework for causal linear token mixing that separates direct input-to-output interactions from recurrent propagation through past outputs. Concretely, it defines a generalized recurrence of the form Y = AX + BY, and argues that different sparsity patterns in A and B recover familiar architectures such as attention and linear recurrence while also suggesting new intermediate designs. The paper then studies structured recurrence patterns that trade decoding/cache complexity against graph-theoretic notions of expressivity, especially shortest information path and congestion, and introduces cache-efficient variants. Empirically, it evaluates these designs on synthetic recall/copy-style tasks and on language modeling, with the main claim being that certain sublinear-complexity constructions recover part of the expressivity of attention while remaining more efficient than dense attention.

**Compliance With Llm Reviewing Policy:**

Affirmed.

**Final Justification:**

The authors additional rebuttal further clarifies my concerns. I raised my score by 1.

**Key Questions For Authors:**

1. The core framework writes the layer as Y = AX + BY, equivalently Y = (I - B)^(-1) A X. Since the composed operator can be absorbed into a single lower-triangular causal matrix, what is the strongest sense in which the A, B decomposition is more than a reparameterization?

2. The paper is framed as a unified view over token mixers, but technically it appears scoped to causal linear token mixing. Can the authors state more explicitly what important classes of architectures are not meaningfully covered, especially selective, nonlinear, hierarchical, or compression-based mechanisms? This would help calibrate the scope of the claims.

3. The discussion of higher-order recurrence references architectures such as log-linear attention. Does the proposed framework actually capture the hierarchical computational structure of such models, or only an equivalent flattened causal operator? Clarifying this would change my assessment of how general and faithful the framework really is.

5. Can the authors comment more on the practical implications of their results compared to existing theoretical results?

**Limitations:**

The limitations are not adequately discussed. The paper should discuss more directly that its framework is limited to causal linear token mixing, and that this restricts its relevance to architectures where nonlinear selectivity, hierarchical computation, or memory compression are central. I would also encourage the authors to be more explicit that the proposed graph-theoretic quantities are proxies for expressivity rather than definitive characterizations of long-context language modeling ability. Societal impacts appear limited and mostly indirect.

**Strengths And Weaknesses:**

**Strengths**

1. The paper addresses a relevant problem: understanding how to trade off efficiency and expressivity in long-context token mixers. I appreciate the attempt to place attention-like and recurrence-like mechanisms into a common view rather than treating them as completely separate families. Even if I am not fully convinced by the depth of the framework, and SSM-like architectures can usually be thought of as linear attentions already, the paper is at least aiming at a meaningful organizing question.

2. I also found the graph-based perspective to be one of the more interesting parts of the paper. In particular, the use of shortest information path and congestion gives a somewhat principled language for explaining why certain sparse recurrence patterns may behave better than naive local recurrence. This part helped the paper feel more analytical than purely architectural.

3. Presentation is generally solid. The central framework is clearly stated, the paper is fairly easy to follow, and the experimental narrative is coherent. I did not have major issues understanding what the authors were trying to do.

**Weaknesses**

1. My main concern is that the core A, B decomposition feels much less fundamental than the paper sometimes suggests. Since Y = (I - B)^(-1) A X, the resulting operator from X to Y can always be absorbed into a single lower-triangular causal matrix M. In that sense, the framework is not a deep characterization of sequence models, but a factorization of a restricted class of causal linear operators. The paper itself notes connections to standard control-theoretic formulations and related prior views for linear attention and SSMs, which further weakens the novelty claim in my view.

2. Related to the weakness above, the framework seems substantially narrower than the framing implies. The paper is explicitly about causal linear token mixing. What the paper proposes is covers attention, linear attention, and some SSM-style operators, but it is not a general theory of long-context sequence modeling. It does not appear to naturally capture architectures where the key ingredient is selective or nonlinear state formation, hierarchical computation, compression-based memory, or routing. I think this scope limitation should be stated much more explicitly.

3. A related issue is that representational inclusion is not the same as faithfully modeling architectural structure. For example, the discussion of higher-order recurrence cites architectures such as log-linear attention, but it is not clear to me that the framework captures the actual hierarchical computational mechanism of such models in a meaningful way, as opposed to only reproducing an equivalent flattened causal operator. That distinction matters if the paper wants to claim a genuinely unifying perspective.

4. Regarding soundness, I did not identify an obvious flaw in the algebraic setup, and the path-length/congestion analysis seems internally reasonable for the graph families considered. However, I was less convinced by the broader “trade complexity for expressivity” message. The graph-theoretic quantities are informative, but they are still proxies, and the paper does not fully establish that they correspond to the practically relevant notion of expressivity for language modeling beyond the selected settings.

5. And regarding significance,  I see some value in the structured constructions and the graph-based viewpoint, and I can imagine some researchers building on those pieces. But because the main decomposition is not especially deep or unique, and because the framework applies to a restricted family of models, I am less convinced that this paper will substantially shift how the field thinks about long-context modeling.

6. For originality, the combination of factorization, graph analysis, and constructions is nontrivial, but the backbone of the paper strikes me more as a tidy reparameterization than a genuinely new theoretical viewpoint. To me, the more original contributions are the particular structured recurrence patterns and the associated analysis, not the A, B framework itself.

7. The related works seems to miss many recent theoretical discussions on long-context sequence modeling, (e.g. https://neurips.cc/virtual/2024/poster/96743 and more recently https://neurips.cc/virtual/2025/loc/san-diego/poster/115721). I'm particularly interested in how the proposed framework is consistent with or improves from the example papers mentioned, but other recent works should also be included and the relations should be discussed more clearly.

---

> ### Author Rebuttal · Authors · 2026-03-31
>
> We thank the reviewer for the detailed feedback and for highlighting important points about the framework’s scope and relevance.
>
> ## Relevance of the framework
>
> ### W1,W5,W6,Q1: The core A, B decomposition feels less fundamental than suggested
>
> We agree that the framework is simply a tidy reparameterization of causal linear recurrence. Such a formulation of linear recurrence has appeared in various forms in the literature, and is especially well-known in control theory. We will reframe the paper to make this point clearer.
>
> Nevertheless, casting standard token mixing operations such as attention and SSMs in this framework provides a unified view and allows systematic analysis of their properties. It also naturally extends to more general models with higher order recurrences.
>
> **Motivation**
>
> A key motivation for the decomposition of the operator into matrices A and B is that while the full operator $(I - B)^{-1} A$ can be dense, the matrices A and B themselves can be sparse and structured.
>
> Taking the example of (diagonal) SSMs, the matrix B is typically strictly lower triangular with nonzero entries only on the first subdiagonal. The Mamba2 paper considers the full dense operator $(I-B)^{-1}$ with semiseparable structure -- we propose to consider $B$ instead, which feels more natural.
>
> **Beyond first-order recurrences**
>
> Allowing $B$ to have nonzero entries on multiple subdiagonals enables higher-order recurrences, which SSMs and linear attention models cannot implement. As pointed out by the reviewer, our main contributions lie in leveraging this framework with sparse patterns in $A$ and $B$, which has not been explored previously.
>
> **Graph interpretation**
>
> A and B can be seen as adjacency matrices defining a directed graph over the sequence, where there is an edge from node j to node i if $\alpha_{i,j}$ or $\beta_{i,j}$ is nonzero. This graph interpretation allows us to analyze the properties of the model in terms of graph-theoretic quantities such as shortest path distance and congestion, which are proxies for the model's ability to propagate information across the sequence.
>
> ### W2,Q2: The framework seems narrower than the framing implies
>
> We acknowledge that some claims about the generality of the framework were too strong (e.g. L.63: "captures attention, SSMs, and their hybrids as special cases."). We will rephrase these claims to be more accurate.
>
> The framework is not meant to capture all sequence modeling architectures. Its focus is to interpolate between attention-based and recurrent models (particularly SSMs) and to analyze information propagation patterns in higher-order linear recurrences.
>
> Selective mechanisms are captured, since coefficients of $A$ and $B$ can be gating functions of inputs. However, nonlinear, hierarchical, or compression-based mechanisms are not included, though extensions may be possible.
>
>
> ### W3,Q3: Representational inclusion vs faithful architectural modeling
>
> This is a very good point, and indeed the framework is only truly meaningful for some specific models such as attention and SSMs. Any causal linear recurrence $Y = MX$ could be expressed by setting $A = M$ and $B = 0$, but this does not capture the architectural structure of all such models.
>
> We focus on attention and SSMs because they are widely used and have a clear representation in our framework. For models like log-linear attention, as the reviewer points out, representation may be less meaningful. We will rephrase our claims to clarify this.
>
> ## W4: Graph-theoretic metrics as proxies for expressivity
>
> We agree that these metrics are proxies. Expressivity in language modeling is multifaceted and difficult to capture with a single metric. We chose shortest path distance and congestion as intuitive, interpretable measures directly linked to the model’s ability to memorize information across the sequence, a key aspect of expressivity.
>
>
> ## W7,Q4: Related work on long-context sequence modeling
>
> - Cirone et al. (2024) analyze SSMs via Linear Controlled Differential Equations. While they have different motivations and consider continuous time, Eq. 5 in their paper is actually quite similar to our framework, and could be seen as a higher-order linear recurrence. Their tools may apply to our framework as well.
> - Chen et al. (2025) define a mutual information based criterion to assess whether a model can capture long-term dependencies: the model's capacity must grow faster than $L^\beta$ where $L$ is the sequence length and $\beta$ is a data-dependent constant. In our paper we derive congestion bounds, similarly to how Chen et al. identify the congestion of SSMs as a bottleneck.
> An interesting corollary is that for the k-th order recurrence, the capacity does not grow with the sequence length, which suggests that such models will always struggle as the sequence length increases.
>
> ---
>
> We hope these clarifications address concerns and better convey the contributions and limitations of our framework.

---

> > ### Author Rebuttal · Reviewer_AXta · 2026-04-02
> >
> > I thank the authors for the rebuttal and appreciate the author for conceding certain points, but the rebuttal mostly confirm my concerns rather than resolving them. If the A, B decomposition is, in the authors' own words, "simply a tidy reparameterization," then the paper's theoretical contribution has to stand on the structured patterns and graph analysis alone, which I found interesting but not sufficient to carry the paper. The fact that the framework is "only truly meaningful" for attention and SSMs is a problem, because the higher-order recurrence direction, which is arguably what makes the paper distinct, ends up without a solid connection to architectures people actually use. I also still do not see enough evidence that shortest path and congestion correspond to expressivity in any meaningful sense beyond the synthetic tasks, which were more or less designed to match these definitions. The link to Chen et al.'s capacity analysis seems interesting and probably worth developing further, but addressing these issues properly would probably require reworking the paper's claims, scope, and theoretical groundings. Therefore, I would like to keep my current score, but would not mind if other reviewers think the paper is interesting and worth publishing.

---

> > > ### Author Response · Authors · 2026-04-06
> > >
> > > We thank the reviewer for their feedback, and particularly regarding the novelty and motivation of the framework.
> > >
> > > ---
> > >
> > > > If the A, B decomposition is, in the authors' own words, "simply a tidy reparameterization," then the paper's theoretical contribution has to stand on the structured patterns and graph analysis alone
> > >
> > > The framework's simplicity does not take away its novelty: while we agree similar forms have appeared in the SSM and linear recurrence literature, we have not seen any previous work come up with an expression as concise, clean and interpretable.
> > >
> > > Moreover, as reviewer MDcW pointed out, our framework is the first to include *softmax* attention naturally. Linear attention methods only support it by resorting to an infinite-dimensional kernel, which is not possible to use in practice. This is a fundamental difference with previous works.
> > >
> > >
> > > ---
> > >
> > > > The fact that the framework is "only truly meaningful" for attention and SSMs is a problem, because the higher-order recurrence direction ends up without a solid connection to architectures people actually use.
> > >
> > > We targeted attention and linear recurrences precisely because these are architectures people actually use in LLMs. On one side we have attention, along with its variants such as local attention, dilated attention, BigBird, and Longformer. On the other side stand linear recurrences such as SSMs (e.g. S4, Mamba, Mamba-2) and linear attention variants (e.g. linear attention, RetNet, GLA).
> > >
> > > Our goal is to interpolate between both behaviors by playing with the pattern of the recurrence: order 1 recovers linear recurrences (linear complexity), while infinite order is comparable to attention (quadratic complexity). We acknowledge that the framework has not been thought to encompass other families of token-mixing layers.
> > >
> > > ---
> > >
> > > > I also still do not see enough evidence that shortest path and congestion correspond to expressivity in any meaningful sense beyond the synthetic tasks, which were more or less designed to match these definitions.
> > >
> > > We agree that performance on the synthetic tasks does not necessarily translate to performance on real language modeling tasks. Still, such tasks are regularly used in the literature to validate architectural innovations and theoretical insights. For instance, the performance discrepancy between SSMs and attention on the associative recall task has motivated the Mamba architecture, which introduced a selective gating mechanism to target this bottleneck [1, 2]. Similarly, multi-hop recall has been extensively studied to understand the limitations of current architectures [3, 4]. In particular, it has been linked to entity tracking [4] and reasoning [5], which are key capabilities for LLMs.
> > >
> > > ---
> > >
> > > We will make sure the framework's novelty and differences to the literature are stated more clearly in the final version of the paper, and hope this answers the reviewer's concerns.
> > >
> > > ---
> > >
> > > [1] Gu, Albert and Tri Dao. *"Mamba: Linear-Time Sequence Modeling with Selective State Spaces."* ArXiv abs/2312.00752 (2023).
> > >
> > > [2] Dao, Tri and Gu, Albert. *"Transformers are SSMs: Generalized Models and Efficient Algorithms Through Structured State Space Duality"*. ICML (2024).
> > >
> > > [3] Wang et al. *"Learning Compositional Functions with Transformers from Easy-to-Hard Data"*. PMLR (2025).
> > >
> > > [4] Fagnou et al. *"Chain and Causal Attention for Efficient Entity Tracking"*. EMNLP (2024).
> > >
> > > [5] Mavi et al. *"Multi-hop Question Answering"*. Foundations and Trends in Information Retrieval (2024).

---

### Official Review · Reviewer_be1e · 2026-03-13

**Soundness:** 3
**Presentation:** 3
**Significance:** 3
**Originality:** 3
**Overall Recommendation:** 4
**Confidence:** 2

**Summary:**

The paper proposes a general framework for analyzing the causal linear token mixing layers, spanning various modern architectures such as attention, SSMs, linear attention, and their hybrids. This framework offers a perspective for designing token-mixing architectures through the tradeoff between computational complexity and expressivity. The paper empirically validates these ideas using both synthetic tasks and language modeling experiments on OpenWebText, including models at 44M and 355M scale.

**Compliance With Llm Reviewing Policy:**

Affirmed.

**Final Justification:**

The rebuttal has fully resolved my concerns.

**Key Questions For Authors:**

In table 1, the $f(k) = k^2 + 1$ model has both constant congestion and constant shortest-path distance between tokens, yet its performance on recall tasks remains relatively weak. This suggests that reachability alone may not be sufficient to explain recall ability. Are there additional metrics, beyond shortest path and congestion, that could better characterize the capacity required for recall tasks?

**Limitations:**

yes

**Strengths And Weaknesses:**

Strengths
- The proposed framework is novel and interesting
- The paper is generally well-written and clear

Weaknesses
- Current experiments only serve as a stress test for the theoretical predictions, leaving the effectiveness on a larger scale unknown

---

> ### Author Rebuttal · Authors · 2026-03-31
>
> We thank the reviewer for their thoughtful feedback and for highlighting both the strengths of our framework and the key concern regarding its effectiveness at scale.
>
> ## W1: Unknown effectiveness at a larger scale
>
> We are aware that the current scale of our experiments is limited. The synthetic tasks only aim at validating our theoretical predictions, while the language modeling experiments show promising results in relatively small models.
>
> We have now run additional experiments to more accurately assess the scaling laws of our proposed models. We have trained 125M and 775M parameter models (in addition to the 355M model in the original submission). The results are reported in the table below, and we will include them in the final version of the paper.
>
> We show the perplexity of the attention baseline, and the relative difference in perplexity for each of the other models:
> | **Layer** | **125M** *(new)* | **355M** | **775M** *(new)* |
> | --- | --- | --- | --- |
> ||*Attention-based*|||
> | Attention | 23.34 | 17.28 | 15.40 |
> | Local attention | +12.10 | +9.21 | +7.81 |
> ||*Our framework*|||
> | Dense $A$ and $B$ | -0.25 | -0.19 | -0.20 |
> | Square pattern | +4.58 | +2.97 | +2.11 |
> | + cache-efficient | +5.45 | +3.51 | +2.98 |
> | Exponential pattern | +5.58 | +3.50 | +2.95 |
> | + cache-efficient | +5.89 | +3.77 | +3.22 |
> | k-th order recurrence (k=8) | +5.99 | +3.75 | +3.15 |
> | First order recurrence | +6.93 | +4.66 | +3.94 |
>
> Note: all models were trained on OpenWebText as in the paper, using the Chinchilla scaling laws. The context length is 1024 for the 125M models and 2048 for the 355M and 775M models.
>
> The Pareto frontier of the models remains consistent across scales, with the general model performing best, followed by the square pattern and exponential pattern models, and finally the recurrent models. While the gap between the attention baseline and the other models decreases as the model size increases, the relative performance of the models remains consistent, suggesting that the tradeoffs we identified do persist at larger scales.
>
> ## Q1: The square pattern is weaker than expected on recall tasks
>
> > In table 1, the $f(k) = k^2+1$ model has both constant congestion and constant shortest-path distance between tokens, yet its performance on recall tasks remains relatively weak. This suggests that reachability alone may not be sufficient to explain recall ability. Are there additional metrics, beyond shortest path and congestion, that could better characterize the capacity required for recall tasks?
>
> We also noted this weaker performance of the square pattern in the recall tasks, which is somewhat surprising given its favorable theoretical properties. We hypothesize that this could be due to the fact that, for this specific pattern,  the shortest path between tokens is not always easy to find.
>
> Indeed, we show the square pattern has a constant shortest path distance between tokens, using Lagrange's four-square theorem. However, this does not necessarily mean that the model can easily find this path during training, especially since the model has to learn to use the right edges in the graph to propagate information across the sequence.
>
> We are still investigating this question, and it is hard to say for sure whether the issue is theoretical (missing metrics) or practical (the model struggles to learn to use the right paths). Practical solutions may involve adding additional inductive biases to the model to help it learn to use the right paths, or using some sort of pretraining task.
>
> Metrics that could potentially be relevant include:
> - the number of paths between tokens (not just the shortest path)
> - the average path length between tokens
> - the sensitivity of the model to perturbations in the paths (i.e. how much does the performance drop if we remove some edges from the graph). This in particular may help modeling training difficulties, since if the model relies on a very specific path to propagate information, it may be more sensitive to perturbations and thus harder to train.
>
> We plan in particular to investigate training dynamics in gradient descent, maybe using tools from optimization theory, to better understand the training difficulties of the square pattern.
>
> ---
>
> We hope that these additional large-scale results and clarifications address the reviewer’s concerns and further support the relevance and robustness of our framework.

---

> > ### Author Rebuttal · Reviewer_be1e · 2026-04-02
> >
> > Thanks for the detailed response. I have revised my score toward acceptance. The expressivity vs training dynamics comparison is very interesting and it reminds me some related discussion on Chain-of-Thoughts:
> >
> > [1] Wen, K., Zhang, H., Lin, H., & Zhang, J. (2024). From sparse dependence to sparse attention: unveiling how chain-of-thought enhances transformer sample efficiency. arXiv preprint arXiv:2410.05459.
> >
> > [2] Kim, J., & Suzuki, T. (2024). Transformers provably solve parity efficiently with chain of thought. arXiv preprint arXiv:2410.08633.

---

> > > ### Author Response · Authors · 2026-04-06
> > >
> > > Thank you for pointing us to these connections. We believe our findings are indeed closely related: Wen et al. argue that CoT helps not only through expressivity, but also by improving sample efficiency via sparse sequential dependencies and sparser attention, while Kim and Suzuki show that explicit intermediate supervision and self-consistency make the relevant multi-step computation much easier to optimize.
> > >
> > > These works provide theoretical and practical insights into how expressivity does not always directly translate to performance, and how training dynamics can be a bottleneck. This is very much in line with our observations on the square pattern, which has favorable theoretical properties but does not fully match those expectations in practice.
> > >
> > > Wen et al. additionally highlight the sparsity of the attention patterns that arise with CoT, which is particularly relevant to our framework since we focus on sparse patterns in A and B.
> > >
> > > We will include a discussion of this literature in the final version of the paper, and thank the reviewer for bringing it to our attention.
> > >
> > > ---
> > >
> > > If you feel that our rebuttal and the new experimental evidence have adequately resolved your concerns, we would sincerely appreciate your consideration in adjusting your score toward acceptance.

---

### Official Review · Reviewer_DAge · 2026-03-13

**Soundness:** 3
**Presentation:** 3
**Significance:** 3
**Originality:** 3
**Overall Recommendation:** 5
**Confidence:** 4

**Summary:**

The paper proposes a unifying view of causal token mixing layers, where the flow of information from inputs and outputs are decomposed into 1) direct input on output influence and 2) recurrent propagation through past outputs. The authors show how this view includes attention, SSMs, and linear attention, and explains how complexity, cache size, and long-range capacity are traded off. Through this framework, authors define concepts  of shortest path and congestion for retrieval and copy tasks, respectively. The authors propose structured sparse time-invariant dependency patterns that can result in various compute complexity classes per step, such as logarithmic and square‑root. They also propose cache-efficient pattern with time-variant sparsity structures and show their periodic structure.  The paper provides empirical study on synthetic recall tasks as well as language modeling, and identify complexity and expressivity trade-offs.

**Compliance With Llm Reviewing Policy:**

Affirmed.

**Final Justification:**

I recommend acceptance. The paper is clearly written and mathematically rigorous, and it introduces an insightful perspective on multi-step dependencies that connects recurrence and attention in a clean way. The experimental design, especially the use of a single backbone to isolate token-mixing effects, is strong. While I had several concerns about clarity and missing distinctions in the definitions and model descriptions, these are largely presentation issues that can be addressed with minor revisions.

**Key Questions For Authors:**

**Questions:**

1. Related to point 2 above, in the definition of communication graph, are inputs and outputs combined into single nodes at each time point?
2. What type of input-dependency is enforced on A and B in the empirical evaluations? And related to point 3 above, how does this affect parallel training?
3. Are there any practical efficiency measures like throughput, latency and GPU memory usage, that could be shared from the language modeling experiments to compare different token mixing variants?

**Limitations:**

Yes

**Strengths And Weaknesses:**

**Strengths:**

The paper is well-written and well-organized. The sections follow smoothly, and diagrams help with visual understanding of the concepts. The paper also provides clean and rigorous mathematical expressions with clear and well defined notations.

The proposed view of multi step dependence between past inputs and outputs is very interesting, and connects recurrence and attention from a different, yet complementary view to that of linearizing attention with kernel trick. Eq. 2 nicely demonstrates the two streams of information, and the graph perspective along with the provided analysis is insightful.

I also appreciated that authors used a single backbone (GPT2+RoPE) to compare all token mixing layers. This nicely isolates only the effect of different design choices. For example, poor performance of diagonal SSMs in recall intensive tasks would not be observed if original architectures were used, as the short convolution specifically help with some of these tasks and mask memory bottlenecks.

**Weaknesses:**

1. One weakness of the proposed framework is that it is oblivious to the channel axis of inputs and outpus, as well as the specific form of input-dependency in A and B. There are important factors such as state expansion, channel mixing within token mixing layers (e.g. non-diagonal SSMs) which drive significant expressivity in models (e.g. delta rule), etc. However, I don’t think this is an issue, as the framework focuses on information propagation across time, but a discussion on this in the main text would be absolutely beneficial.
2. The distinction between patterns of A and B is missing and leads to confusion. For example, from definition 4.1 (eq. 3), $\alpha_{i,j} \neq 0$ “or” $\beta_{i,j} \neq 0$ implies the overall pattern hold if the union of A and B includes the said pattern f. Or definition 4.3 of the communication graph would benefit from distinction between two types of nodes (input and output). It is not clear that DAG is defined over only inputs, only outputs, or both at the same time. This causes some confusions later when shortest path is discussed, as the shortest path should be defined over paths “from input node j to state node i”.
3. There is not much information on the specific input-dependency form of A and B. Appendix only states it uses attention, but it is not clear how attention over past states is implemented, as it could mean B depends on itself, which makes parallelizing over time impossible or perhaps inefficient.
4. I could not quite easily understand how cache efficient variants are defined, and perhaps a more clear explanation is needed. Also, in Definition 4.10 Eq. 9, how can $\alpha_{i,j}$ be nonzero when j>i (causality)
5. Some models in Table 1 are not properly defined. For example, it’s not clear what local recurrence is. Also “General” would benefit from more explanation or a better term.
6. I think using different (specifically complementary) patterns for A and B could result in more interesting models since the flow of information from past inputs and past states could be disentangled, but this is not considered at all.

---

> ### Author Rebuttal · Authors · 2026-03-31
>
> We thank the reviewer for the positive feedback and insightful questions.
>
> ## W1.a: Missing discussion of the role of dimension
>
> We fully agree that dimension should be discussed more extensively. As the reviewer points out, our framework -- and the theoretical insights we provide -- focus on the temporal dimension and how information is propagated along the sequence. The actual amount of information that can be stored and propagated by the model (i.e. the actual "memory capacity") depends linearly on the channel dimension, and thus the channel dimension can be leveraged to increase the capacity of the model and compensate for the higher congestion. In practice, many models use much larger channel dimensions (e.g. SSMs using state-expansion) to be competitive with attention-based models while remaining linear in the sequence length.
>
> Regarding channel mixing as in non-diagonal SSMs, we agree that the framework does not explicitly capture this aspect. One possible extension would be to allow $\alpha_{i,j}$ and $\beta_{i,j}$ to be matrices instead of scalars, enabling channel mixing. However, this would significantly complicate the theoretical analysis and may make it much more complex overall.
>
> # Input-dependency in A and B
>
> ### W1.b: Missing discussion
>
> The framework is deliberately agnostic to the specific parametrization of A and B, since we want it to be as general as possible and include a wide range of models. However, we agree that the specific input-dependency form of A and B can have a significant impact on the performance of the model. The theoretical results could also be further refined by including additional assumptions on the parametrization of A and B. For instance, linear attention and the delta rule can benefit from linear algebra theory.
>
> ### W3,Q2: Specific input-dependency form of A and B
>
> The only assumption we make is that the coefficients of A and B are functions of the input tokens. They should all be precomputable before processing the sequence, thus allowing for efficient implementation -- we will clarify this point in the paper. In our experiments, A and B are computed using attention matrices between keys and queries that are functions of the input tokens only.
>
> ## Clarifications
>
> ### W2,Q1: The distinction between patterns of A and B is missing and leads to confusion
>
> In the theoretical section, we actually make the assumption that A and B have the same pattern (except that A can have a nonzero diagonal). Thus the edges of the graph do not need to separately indicate whether they come from A or B, and we do not need to distinguish between input nodes and output nodes. Inputs and outputs are indeed combined into single nodes at each time point. We will reformulate the theoretical section to make this point clearer.
>
> ### W4: I could not quite easily understand how cache efficient variants are defined
> We understand that the definition of cache-efficient variants is not entirely clear, especially since Definition 4.10 is quite abstract. The idea is that when the model tries to pay attention an index $j = i-f(k)$ (as in the usual pattern from Definition 4.1), we replace $j$ by $j' \geq j$ which is the closest index to $j$ that is in the cache. This way, the model only attends elements that are in the cache.
>
> This is a recursive definition, but we also derived the closed form solution in Proposition 4.12.
>
> ### W4.b: Also, in Definition 4.10 Eq. 9, how can $\alpha_{i,j}$ be nonzero when j>i (causality)
>
> Indeed, this is an error, $i$ and $j$ were swapped. The equation should read $1 \leq j \leq i \leq n$ instead, we will fix this in the paper.
>
> ### W5: Some models in Table 1 are not properly defined
> Indeed, the terms are vague and must be clarified. In addition to better definitions in the caption, we could replace "local recurrence" by "k-th order recurrence" and "general" by "dense recurrence" for instance.
>
> ## W6: Using different (specifically complementary) patterns for A and B could result in more interesting models
>
> This is a very interesting research direction which we aim to explore in the future. In the current paper, we focused on models where A and B have the same pattern, which simplifies both theory and empirical evaluation. However, we agree that using different patterns for A and B could definitely lead to more powerful models.
>
> ## Q3: Are there any practical efficiency measures like throughput, latency and GPU memory usage
>
> We have not measured these metrics in our current experiments since they would require a highly optimized implementation of the models, which we currently do not have.
>
> While we provide the time complexities and cache sizes of the models in the paper, we agree that actual measurements would be very informative.
>
> We are currently developing optimized decoding algorithms leveraging cache-efficient variants and plan to report such metrics in the final version.
>
> ---
>
> We hope these clarifications address the concerns and improve the clarity of the paper.

---

> > ### Author Rebuttal · Reviewer_DAge · 2026-03-31
> >
> > Thank you for your detailed response. I have already recommended acceptance and will keep my score as such.

---

### Official Review · Reviewer_MDcW · 2026-03-18

**Soundness:** 3
**Presentation:** 3
**Significance:** 2
**Originality:** 2
**Overall Recommendation:** 4
**Confidence:** 3

**Summary:**

This paper proposes a factorisation for causal linear token mixing in the form $Y=(I-B)^{-1}AX$, showing that such mixers can be decomposed into direct one-step input influence and recurrent propagation through past outputs. Within this factorisation, attention, SSM recurrences, and some hybrids can be expressed in a common form, and the paper derives structured recurrence patterns with controllable complexity–expressivity tradeoffs. The empirical study includes synthetic recall/copy tasks and small OpenWebText language-model pretraining.

**Compliance With Llm Reviewing Policy:**

Affirmed.

**Final Justification:**

The authors have addressed most of my concerns, except for the practical deployment issue, which still appears to require specialized kernels and careful cache design for real-world use. That said, I find the work interesting in terms of both its formulation and empirical results, and I am therefore increasing my score.

**Key Questions For Authors:**

1) The paper positions the framework as covering attention, SSMs, and hybrids. Can the authors include direct empirical comparisons to stronger modern efficient baselines, such as Mamba or gated linear-attention models and explain how they fit in with the proposed framework?

**Limitations:**

yes

**Strengths And Weaknesses:**

Strengths

1. The factorization ($Y=(I-B)^{-1}AX$) provides a useful framework-level view of causal token mixing, where (A) captures direct input–output interactions within a step and (B) governs recurrent information propagation across steps.

2. By varying the structure and sparsity of (A) and (B), the framework spans a broad family of architectures that are often studied separately, while maintaining explicit causality through triangular structure.

Weaknesses

1. The underlying formulation of linear recurrence is not entirely new, as closely related views are already common in control theory and have also appeared in prior work on linear attention and SSM variants. The main novelty seems to lie more in simplifying and extending the framework to softmax-attention.

2. The empirical validation remains somewhat limited. The largest language-modeling experiments use (355M params) transformers trained on OpenWebText, and the paper itself notes that the experiments are small-scale and use unoptimized kernels. As a result, it is unclear whether the reported tradeoffs would persist at larger and more practically relevant scales.

3. Despite the broad framing around attention, SSMs, and hybrids, the empirical comparisons are mostly against full attention, local attention, diagonal SSMs, local recurrence, and variants within the proposed framework, rather than stronger modern efficient baselines such as Mamba or other gated linear-attention models.

4. The systems claim is still somewhat incomplete. While the paper argues for favorable asymptotic complexity regimes, it also acknowledges that realizing actual speedups would require specialized block-triangular kernels and careful cache design.

---

> ### Author Rebuttal · Authors · 2026-03-31
>
> We thank the reviewer for their helpful feedback and for highlighting key concerns on novelty, scale, and baselines.
>
> ## W1: Novelty of the framework
>
> We acknowledge that such formulations of linear recurrence have appeared in various forms in the literature. This is especially well-known in control theory.
>
> However, we distinguish our work from SSM and linear attention works in several ways:
> - the framework includes softmax attention without resorting to kernel approximations.
> - the matrix $B$ is freely parameterized and can be dense strictly lower triangular, enabling more flexible information propagation patterns (i.e. higher order recurrences). SSMs and linear attention models typically are only able to implement first-order recurrences.
>
> Moreover, the core idea is to leverage this framework with sparse patterns in $A$ and $B$, which has not been explored. Even when $A$ and $B$ are sparse, the operator $(I - B)^{-1} A$ can be dense and enables powerful information propagation.
>
>
> ## W2: Experiment scale
>
> We acknowledge that experiments are limited in scale. Synthetic tasks validate theory, while language modeling shows promising results at small scale.
>
> We have now run additional experiments to more accurately assess the scaling laws of our proposed models. We trained 125M and 775M models (in addition to 355M). Results are below and will be included in the final version.
>
> We report attention perplexity and relative differences to attention for other models:
> |**Layer**|**125M** *(new)*|**355M**|**775M** *(new)*|
> |---|---|---|---|
> ||*Attention-based*|||
> |Attention|23.34|17.28|15.40|
> |Local attention|+12.10|+9.21|+7.81|
> ||*Our framework*|||
> |Dense $A$ and $B$|-0.25|-0.19|-0.20|
> |Square pattern|+4.58|+2.97|+2.11|
> |+ cache-efficient|+5.45|+3.51|+2.98|
> |Exponential pattern|+5.58|+3.50|+2.95|
> |+ cache-efficient|+5.89|+3.77|+3.22|
> |k-th order recurrence (k=8)|+5.99|+3.75|+3.15|
> |First order recurrence|+6.93|+4.66|+3.94|
>
> All models are trained on OpenWebText using Chinchilla scaling laws. Context length is 1024 (125M) and 2048 (355M, 775M).
>
> The Pareto frontier remains consistent across scales: the general model performs best, followed by square/exponential patterns, then recurrent models. While gaps to attention shrink with scale, relative performance remains consistent, suggesting the identified tradeoffs persist.
>
>
> ## Other baselines
>
> ### W3,Q1.a: Stronger modern efficient baselines
>
> We agree that it is important to compare our models to stronger modern efficient baselines. However, for the comparison to be fair, we need to strip these models down to their core components, to isolate token-mixing capacity. For instance, Mamba layers typically include a small convolution and nonlinearity, along with state-expansion and head normalization.
>
> We have implemented this stripped-down version of Mamba and trained it in the 125M parameter setting.
> | **Layer** | **Perplexity** |
> | --- | --- |
> | Mamba | 30.64 |
> | First order recurrence (ours) | 31.28 |
>
> For reference, attention stands at 23.34 and the k-th order recurrence (k=8) stands at 29.33 (both from the previous table).
>
> While Mamba seems to perform slightly better, the gap is not very large, and the model is still significantly worse than the other patterns in our framework.
>
> ### Q1.b: How do Mamba and GLA fit in with the proposed framework?
>
> The linear recurrences in Mamba and GLA can be expressed in our framework by appropriately defining the matrices A and B. Note however that these models also include additional components such as convolutions and nonlinearities, which are not captured by our framework.
>
> In Appendices A.3 and A.4, we describe how to express SSMs and linear recurrences in our framework. State-expansion (present in both models) can be seen as sharing the same matrices A and B for different channels.
>
> In the case of Mamba, the element-wise equation for the scalar $h_i$ is:
> $$h_i = \exp(\Delta_i a) h_{i-1} + \Delta_i b_i x_i$$
> where $\Delta_i$ is a small (input-dependent) step size, $a$ is a learnable parameter, and $b_i$ is a function of the input tokens. This can be expressed in our framework by defining $A$ and $B$ as follows:
> $$
> A = \begin{bmatrix}
> \Delta_1 b_1 &&& 0\\\\
>  & \Delta_2 b_2 &&\\\\
>  &  & \ddots &\\\\
> 0 & & & \Delta_n b_n
> \end{bmatrix}
> $$
>
> $$
> B = \begin{bmatrix}
> 0 &&& 0 \\\\
> \exp(\Delta_2 a) &&&\\\\
> & \ddots &&\\\\
> 0 && \exp(\Delta_n a)&0
> \end{bmatrix}
> $$
>
> ## W4: Specialized kernels
>
> We agree that practical deployment requires specialized kernels and careful cache design. We however leave this to future work since the main focus of the paper is to provide theoretical insights and a framework for understanding the tradeoffs between different information propagation patterns, rather than to provide a highly optimized implementation of the models.
>
> ---
>
> We hope these clarifications and additional experiments address the reviewer’s concerns and further demonstrate the relevance, generality, and practical potential of our framework.

---

> > ### Author Rebuttal · Reviewer_MDcW · 2026-04-04
> >
> > Thank you to the authors for addressing my questions and for conducting additional experiments. I will raise my score to 4.

---

### Decision · Program_Chairs · 2026-04-30

**Decision:**

Accept (regular)

**Comment:**

This paper studies causal linear token mixing by decomposing it into two components: direct input influence on outputs and recurrent propagation through past outputs. This formulation is linked to existing linear recurrent models such as linear attention and SSMs. The authors further show that different structural choices of the matrices A and B control the trade-off between complexity and model expressivity.

The paper introduces a unified framework for analyzing such models. While the core formulation is not entirely novel and similar perspectives have appeared in prior work, this paper provides a systematic study of the complexity/expressivity trade-off within this framework. The authors support their claims with both synthetic and language modeling experiments. During the rebuttal and discussion phases, the authors addressed most of the reviewers’ concerns. Several reviewers updated their assessments toward acceptance.